# LEARNING ESCORTED PROTOCOLS FOR MULTISTATE FREE-ENERGY ESTIMATION

**Lars Holdijk**
Department of Computer Science
University of Oxford

**Nithishwer Mouroug Anand**
Department of Biochemistry
University of Oxford

**Michael Bronstein**
Department of Computer Science
University of Oxford,
AITHYRA

**Max Welling**
AMLAB
University of Amsterdam,
CuspAI

## ABSTRACT

Estimating relative free energy differences between multiple thermodynamic states lies at the core of numerous problems in computational biochemistry. Traditional estimators, such as Free Energy Perturbation and its non-equilibrium counterpart based on the Jarzynski equality, rely on defining a switching protocol between thermodynamic states and computing the free energy difference from the work performed during this process. In this work, we present a method for learning such switching protocols within the class of escorted protocols, which combine deterministic and stochastic steps. For this purpose, we use Conditional Flow Matching and introduce Conditional Density Matching (CDM) to estimate changes in free energy. We further reduce the variance in the multi-state setting by coupling multiple flows between thermodynamic states into a flow graph of escorted protocols, enforcing estimator consistency across different transition paths.

## 1 INTRODUCTION

In recent years there has been significant interest in the application of machine learning to problems in computational biochemistry, such as protein folding (Jumper et al., 2021; Abramson et al., 2024; Bose et al., 2024) and molecular conformer generation (Gómez-Bombarelli et al., 2018) for drug discovery (Wan et al., 2022) and materials design (Merchant et al., 2023). While initial approaches focused on generating single static examples of systems of interest, recent efforts have shifted towards generating the full dynamic ensemble of the system (Noé et al., 2019; Holdijk et al., 2023; Tan et al., 2025; Akhound-Sadegh et al., 2024). This shift towards the entire dynamical ensemble has opened up the possibility of using these methods in computational biochemistry beyond what generating static samples allows. Notable among these is the problem of estimating free-energy differences between different thermodynamic states, a crucial aspect of many tasks in this field, such as binding affinity prediction (Mobley and Gilson, 2017) and other components of the drug discovery pipeline (Cournia and Chipot, 2024). Traditional estimators such as multiwindow Free-Energy Perturbation (FEP) (Zwanzig, 1955; Wang et al., 2015) and its non-equilibrium variants, defined through the Jarzynski equality (Jarzynski, 1997), are among the most commonly used for this purpose.

In this work, we specifically consider the class of Escorted Non-EQuilibrium (E-NEQ) estimators (Vaikuntanathan and Jarzynski, 2008; 2011) based on the aforementioned Jarzynski equality. These estimators combine dynamics that preserve a given stationary distribution $p(x, t)$, such as Langevin dynamics, with a deterministic escorting vector $b(x, t)$ field to transition samples between thermodynamic states. To minimize variance in the finite-sample limit, the preserved stationary distribution and the escorting vector field must together satisfy the continuity equation $\frac{\partial p(x,t)}{\partial t} + \nabla \cdot (p(x,t)b(x,t)) = 0$ (Zhong and DeWeese, 2024). Traditionally, finding such a pair is non-trivial.

To overcome this barrier, we propose learning the stationary distribution and escorting vector field jointly by extending the Conditional Flow Matching (CFM) framework (Lipman et al., 2023) with an additional Conditional Density Matching (CDM) objective (Sec. 3). We hypothesize that learning the stationary distribution and driving force in a unified framework will allow them to compensate for each other's small errors, leading to a more accurate estimator than using only vector fields,

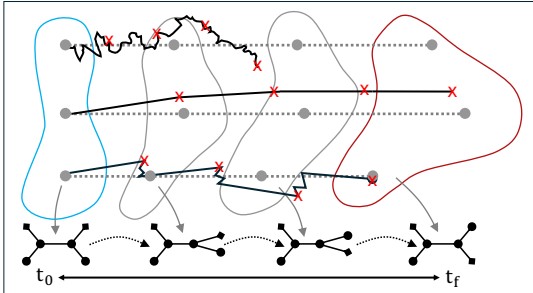 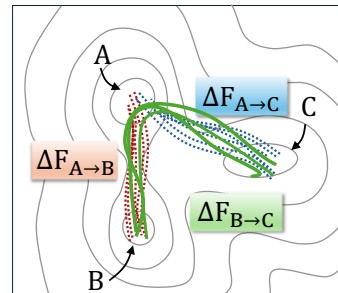

Figure 1: *(a)* Illustration of processes transforming between two thermodynamic states (blue and red) of the same molecular system. Using our proposed Conditional Density Matching framework we learn approximate intermediate densities (grey) along which the learned flow (grey dotted lines) transports samples. The top stochastic line shows how stochastic processes purely targeting the intermediate distributions will not fully reach the target state, while the middle process shows that, due to integration error, deterministic processes may deviate from the true path. Combining stochastic and deterministic processes allows the two approaches to compensate, resulting in more accurate sampling. *(b)* Illustration of combining two free-energy estimates with a central connecting node (A) to obtain the free-energy $\Delta F_{B \to C}$ difference between the nodes not connected.

such as done in methods such as Targeted Free-Energy Perturbation (TFEP) (Jarzynski, 2002) as illustrated in Fig. 1. To further improve the applicability of the framework, we propose two important practical considerations in the form of Lie-Trotter splitting to reduce the computational cost of the work calculation (Sec. 4.1) and a flow graph construction to reduce the combinatorial complexity of learning protocols for multi-state free-energy estimation (Sec. 4.2).

The remainder of our paper is organized as follows to introduce the core contributions of our work:

- Sec. 2 introduces the theory of non-equilibrium free-energy estimation, focusing on topics such as the Jarzynski equality, escorted switching protocols, and bi-directional estimators.
- Sec. 3 then relates escorted switching protocols with the Flow Matching framework and introduces one of our core contributions in the form of conditional Density Matching. Collectively these form the learning objectives for our proposed Escorted Non-EQuilibrium (E-NEQ) estimator.
- To improve the efficiency of multistate free-energy estimation, Sec. 4.1 proposes a Lie–Trotter splitting scheme and Sec. 4.2 proposes the concept of *Escorted Protocol Flow Graphs* to combine multiple trained escorted switching protocols to reduce training requirements.
- In Sec. 5 the E-NEQ estimator is experimentally validated using the well-known Alanine Dipeptide (ADP) system. ADP is an important benchmark due to its multistate free-energy surface.

## 2 BACKGROUND

As stated, in this work we are interested in estimating the relative free-energy differences $\Delta F_{A \to B}$ between pairs of thermodynamic states $A$ and $B$. Each thermodynamic state $A$ is associated with a potential energy function $U_A : \mathbb{R}^{3N} \to \mathbb{R}$, which defines the distribution of possible microscopic states $x \in \mathbb{R}^{3N}$ through the Boltzmann distribution

$$p_A(x) = \frac{1}{Z_A} \exp\left(-\beta U_A(x)\right), \quad Z_A = \int_{\mathbb{R}^{3N}} \exp\left(-\beta U_A(x)\right) dx. \tag{1}$$

Generally, $\beta = \frac{1}{k_B T}$, with $k_B$ representing the Boltzmann constant and $T$ the temperature of the system in Kelvin. Using the partition function $Z_A$, the equilibrium free-energy of state $A$ can be defined as $F_A = -\beta^{-1} \ln Z_A$ and their relative free-energy difference as

$$\Delta F_{A \to B} = F_B - F_A = -\beta^{-1} \ln \frac{Z_B}{Z_A}. \tag{2}$$

As exemplified by the definition of the partition function, estimating free energies by directly integrating over the entire space $\mathbb{R}^{3N}$ is prohibitively expensive. As such, various fields have developed methods to estimate this quantity more efficiently.

### 2.1 NON-EQUILIBRIUM FREE-ENERGY ESTIMATION

We will specifically focus on the class of *alchemical Non-EQuilibrium* (NEQ) free-energy estimation methods, which are a subset of the class of path-based free-energy estimation methods. In path-based free-energy estimation, a central object of in is the switching protocol $U_{A \to B}$.

**Definition 2.1** (Switching Protocol). Given two thermodynamic states $A$ and $B$ with potential energy functions $U_A$ and $U_B$, a **switching protocol** $U_{A\to B} : \mathbb{R}^{3N} \times [0, t_f] \to \mathbb{R}$ is a time-dependent potential energy function with boundary conditions $U_{A\to B}(x, 0) = U_A(x)$ and $U_{A\to B}(x, t_f) = U_B(x)$.

Using such a switching protocol $U_{A\to B}$, a system $x$ initially equilibrated with respect to the potential energy function $U_A$, can be driven from state $A$ to state $B$ according to the time-dependent dynamics

$$\partial_t \rho(x, t) = \mathcal{L}_t \rho(x, t), \qquad \rho(x, 0) = p_A(x), \tag{3}$$

where $\mathcal{L}_t$ is a time-dependent forward (Fokker–Planck/Liouville) generator, specified so that for all $t$ the target distribution $p(x, t) \propto \exp\left(-\beta U_{A\to B}(x, t)\right)$ is stationary for the frozen-time dynamics, i.e. $\mathcal{L}_t p(\cdot, t) = 0$. Notably, it does not have to be the case that the system is in equilibrium at all times, i.e. $\rho(x, t) = p(x, t)$ does not need to hold for all $t$.

Under such dynamics defined by the protocol $U_{A\to B}$, the second law of thermodynamics tells us that, with thermodynamic work defined as

$$W_{A\to B}(\mathbf{x}) = \int_0^{t_f} \frac{\partial U_{A\to B}(x_t, t)}{\partial t} \, \mathrm{d}t, \tag{4}$$

the average work performed on the system along trajectories $\mathbf{x}$ generated by the dynamics in Eq. (3) is an upper bound on the free-energy difference between the two states

$$\Delta F_{A\to B} \leq \langle W_{A\to B}(\mathbf{x}) \rangle_{U_{A\to B}}. \tag{5}$$

where the shorthand $\langle \cdot \rangle_{U_{A\to B}}$ denotes the ensemble average over trajectories $\mathbf{x}$ generated by the dynamics in Eq. (3). Crucially, equality only holds here for quasi-static (infinitely slow, $t_f = \infty$) processes where the system remains in equilibrium at all times, i.e. $\rho(x, t) = p(x, t)$ for all $t$.

### 2.1.1 JARZYNSKI EQUALITY

For any such switching protocols, Jarzynski (Jarzynski, 1997) showed the remarkable result that instead of restricting to quasi-static processes to obtain an equality for the thermodynamic work we can consider the ensemble average of the exponential of the work.

**Theorem 2.2** (Jarzynski Equality (JE) (Jarzynski, 1997)). *Given switching protocol $U_{A\to B}$ and time-dependent dynamics with frozen time stationary distribution $p(x, t) \propto \exp\left(-\beta U_{A\to B}(x, t)\right)$ for all $t \in [0, t_f]$ as in Eq. (3), we have that along trajectories $\mathbf{x}$ generated by the dynamics*

$$\left\langle e^{-\beta W_{A\to B}(\mathbf{x})} \right\rangle_{U_{A\to B}} = e^{-\beta \Delta F_{A\to B}}. \tag{6}$$

Important to note here is that the class of dynamical processes considered in the JE only require stationarity. The JE does not depend on ergodicity and is therefore valid for a wide range of dynamics including both time-dependent stochastic dynamics such as underdamped and overdamped Langevin as well as deterministic dynamics, e.g. Hamiltonian. Extra care is needed when momenta are considered (e.g., underdamped Langevin dynamics). In this case, a state-dependent Hamiltonian, rather than the potential energy, should be used in the definition of work in Eq. (4).

Using standard Monte Carlo integration with trajectories $\mathbf{x}_n$ generated according to Eq. (3), the JE provides a consistent but generally biased estimator of the $\Delta \mathcal{F}_{A\to B}$ in the finite-sample setting:

$$\Delta F_{A\to B} \approx \Delta \mathcal{F}_{A\to B} = -\beta^{-1} \ln\left(\frac{1}{N} \sum_{n=1}^{N} e^{-\beta W_{A\to B}(\mathbf{x}_n)}\right), \tag{7}$$

**JE Estimator Variance** Both the bias and the variance of this estimator grow with the excess work $W^{\mathrm{ex}} = \langle W \rangle_{U_{A\to B}} - \Delta F_{A\to B}$ (Geiger and Dellago, 2010; Gore et al., 2003), which for every time $t$ is lower bounded by the Kullback–Leibler divergence between the two distributions:

$$W_t^{\mathrm{ex}} \geq \beta^{-1} D_{\mathrm{KL}}(\rho(x, t) || p(x, t)), \quad \forall t \in [0, t_f]. \tag{8}$$

As such, intuitively, while the JE is valid for any switching protocol the variance and the bias of the estimator are determined by how much the instantaneous distribution $\rho(x, t)$ *lags* behind the stationary distribution $p(x, t)$ at every time $t$ defined by the switching protocol $U_{A\to B}$.

### 2.1.2 ESCORTED JARZYNSKI EQUALITY

To reduce the amount of lag between the instantaneous distribution $\rho(x, t)$ and the target distribution $p(x, t)$ Vaikuntanathan and Jarzynski (2008) introduced the concept of escorted switching protocols.

> **Definition 2.3** (Escorted switching protocol)**.** Given two thermodynamic states $A$ and $B$ with potential energy functions $U_A$ and $U_B$, an **escorted switching protocol** $(U_{A \to B}, b)$ consists of a time-dependent potential $U_{A \to B} : \mathbb{R}^{3N} \times [0, t_f] \to \mathbb{R}$ and a time-dependent vector field $b : \mathbb{R}^{3N} \times [0, t_f] \to \mathbb{R}^{3N}$ with boundary conditions $U_{A \to B}(x, 0) = U_A(x)$ and $U_{A \to B}(x, t_f) = U_B(x)$ and $b(x, 0) = b(x, t_f) = 0$.

Given an escorted switching protocol, the escorted *dynamics* driving a system from state $A$ to state $B$ adds deterministic advection based on the time-dependent vector field $b$

$$\partial_t \hat{\rho}(x, t) = \widehat{\mathcal{L}}_t \hat{\rho}(x, t) = \mathcal{L}_t \hat{\rho}(x, t) - \nabla \cdot \big(b(x, t) \hat{\rho}(x, t)\big), \qquad \hat{\rho}(x, 0) = p_A(x). \tag{9}$$

Here $\mathcal{L}_t$ denotes the forward operator that satisfies frozen time stationarity $\mathcal{L}_t p(\cdot, t) = 0$ for $p(x, t) \propto \exp(-\beta U_{A \to B}(x, t))$, as defined in Eq. (3). Under escorted dynamics, Vaikuntanathan and Jarzynski (2008) showed that with an alternative definition of the work, the JE still holds.

> **Theorem 2.4** (Escorted Jarzynski Equality (E-JE) Vaikuntanathan and Jarzynski (2008))**.** *Given an escorted switching protocol $(U_{A \to B}, b)$ and time-dependent dynamics as in Eq. (9), we have that along trajectories $\mathbf{x}$ generated by the dynamics*
>
> $$\left\langle e^{-\beta \hat{W}_{A \to B}(\mathbf{x})} \right\rangle_{(U_{A \to B}, b)} = e^{-\beta \Delta F_{A \to B}}. \tag{10}$$
>
> *where the escorted work $\hat{W}_{A \to B}(\mathbf{x})$ along trajectory $\mathbf{x}$ is defined as*
>
> $$\hat{W}_{A \to B}(\mathbf{x}) = \int_0^{t_f} \Big( \partial_t U_{A \to B}(x_t, t) + b(x_t, t) \cdot \nabla U_{A \to B}(x_t, t) - \beta^{-1} \nabla \cdot b(x_t, t) \Big) \, \mathrm{d}t. \tag{11}$$

Notably, because $b$ is not involved in enforcing that the dynamics have stationary distribution $p(x, t) \propto \exp\left(-\beta U_{A \to B}(x, t)\right)$ it can be freely chosen to minimize the amount of lag and with that the variance of the escorted estimator $\Delta \widehat{\mathcal{F}}_{A \to B} = -\beta^{-1} \ln(\frac{1}{N} \sum_{n=1}^N e^{-\beta \hat{W}_{A \to B}(\mathbf{x}_n)})$. Notably among the choices of $b$ are the case where $b = 0$, which is equivalent to the non-escorted case, as well as the following optimal choice of $b$ as shown by Zhong et al. (2023):

> **Theorem 2.5** (Optimal Escorted Switching Protocol (Zhong et al., 2023))**.** *If $b(x, t)$ and $p(x, t) \propto \exp\left(-\beta U_{A \to B}(x, t)\right)$ collectively solve the continuity equation $\partial_t p + \nabla \cdot (p \, b) = 0$, then, for every trajectory $\mathbf{x}$ generated by the dynamics Eq. (9), we have $\hat{W}_{A \to B}(\mathbf{x}) = \Delta F_{A \to B}$.*

Crucially, what this shows is that if we learn $b$ and $U_{A \to B}$ collectively to solve the continuity equation, while maintaining the boundary conditions $U_{A \to B}(x, 0) = U_A(x)$ and $U_{A \to B}(x, t_f) = U_B(x)$, then a single trajectory $\mathbf{x}$ generated by the dynamics in Eq. (9) suffices to estimate the free-energy difference $\Delta F_{A \to B}$. In the remainder of this work we will explore how we can learn these components using Conditional Flow Matching and our proposed Conditional Density Matching.

### 2.1.3 BI-DIRECTIONAL SAMPLING AND THE CROOKS IDENTITY

In addition to introducing the escorting velocity field $b$, a second approach to reduce the variance of our estimator of the free-energy difference is to use bi-directional estimators. Instead of only evolving samples from the initial thermodynamic state $A$, it is equally straightforward to start from state $B$ and define a reverse protocol $U_{B \to A}$ with boundary conditions $U_{B \to A}(x, 0) = U_B(x)$ and $U_{B \to A}(x, t_f) = U_A(x)$. In the context of non-equilibrium free-energy estimation, this idea underpins the Crooks Fluctuation Theorem (CFT) (Crooks, 1998):

$$\frac{P_{A \to B}(W)}{P_{B \to A}(-W)} = \exp[\beta \left(W - \Delta F_{A \to B}\right)]. \tag{12}$$

Here $P_{A \to B}(W)$ is the probability of observing work $W$ for a process starting from $x_0 \sim p_A$ and evolved under the forward dynamics, while $P_{B \to A}(-W)$ is the probability of observing work $-W$

under the time-reversed protocol. This fluctuation theorem holds both for the non-escorted case and the escorted case with $\hat{W}_{A\to B}(\mathbf{x})$ replacing $W(\mathbf{x})$.

For the escorted case, the time-reversed protocol $(U_{B\to A}, \tilde{b})$ is given by $U_{B\to A}(x,t) = U_{A\to B}(x, t_f - t)$ and $\tilde{b}(x,t) = -b(x, t_f - t)$. Consequently the time-reversed escorted dynamics driving the system from state $B$ to state $A$ are given by

$$\partial_t \hat{\rho}_R(x,t) = \tilde{\mathcal{L}}_t \hat{\rho}_R(x,t) - \nabla\cdot\big(\tilde{b}(x,t)\,\hat{\rho}_R(x,t)\big), \qquad \hat{\rho}_R(x,0) = p_B(x). \tag{13}$$

Here $\tilde{\mathcal{L}}_t$ denotes the reverse operator of a non-escorted dynamics that satisfies frozen time stationarity $\tilde{\mathcal{L}}_t \tilde{p}(x,t) = 0$ for $\tilde{p}(x,t) = p(x, t_f - t) \propto \exp(-\beta U_{B\to A}(x,t))$. In the case where dynamics with momenta are considered, the momenta have to be reversed in the reverse generator $\tilde{\mathcal{L}}_t$.

Crucially, the Crooks Fluctuation Theorem expresses $\Delta F_{A\to B}$ as the solution to a single-parameter problem. Rather than directly estimating $\Delta F_{A\to B}$ via the JE, which often suffers from high variance and finite sample bias due to its exponential average, the CFT can be inverted to solve for $\Delta F_{A\to B}$ as the parameter that best fits a given set of observed forward and reverse work for a given protocol.

The remainder of this work will only consider using the CFT instead of the JE. Practically, this is implemented by the Bennett Acceptance Ratio (BAR) estimator (Bennett, 1976).

## 2.2 RELATED WORK

**Traditional Free-Energy Estimation** In addition to the traditional alchemical non-equilibrium free-energy estimation approaches introduced in the previous section, a large collection of other approaches have been proposed. We briefly outline the core lines of work here. Closely related to the approaches considered in this work are alchemical *equilibrium* approaches such as (iterative) Free-Energy Perturbation (FEP) (Zwanzig, 1955) and Thermodynamic Integration (TI) (Kirkwood, 1935), which depend on the Zwanzig equation (Zwanzig, 1955). Next are the *non-alchemical* path-based approaches such as transition path sampling (Bolhuis et al., 2002) and nudged elastic band sampling (Henkelman et al., 2000). Non-alchemical approaches rely on frameworks such as Transition State Theory (Truhlar et al., 1996) and the Arrhenius/Eyring equation (Eyring, 1935).

**Neural Free-Energy Estimation Methods** Within the family of alchemical free-energy estimation approaches, most proposed neural approaches focus on the Targeted Free-Energy Perturbation method (TFEP) (Jarzynski, 2002). The TFEP method can be considered a special case of the escorted switching protocol where only the deterministic vector field $b$ is used to define an invertible mapping. TFEP has been studied within the machine learning context in Wirnsberger et al. (2020); Rizzi et al. (2023; 2021); Erdogan et al. (2024); Zhao and Wang (2023), which all use variants of MLE-trained discrete normalizing flows or Flow Matching. Within the same alchemical family, Máté et al. (2024; 2025) propose a neural version of Thermodynamic Integration. Lastly, most closely related to methods proposed here is the work by He et al. (2025), which also parameterizes an escorted switching protocol. Crucially, their proposed method FEAT uses two different protocols for the forward and backward process between two states and does not consider the multistate setting.

## 3 LEARNING ESCORTED PROTOCOLS USING FLOW AND DENSITY MATCHING

Having discussed the background of (Escorted-) Non-Equilibrium free-energy estimators, we now propose a new method for parameterising and learning the escorted switching protocol $(U_{A\to B}^\theta, b^\phi)$ such that they collectively solve the continuity equation, and adhere to the boundary conditions $U_{A\to B}^\theta(x,0) = U_A(x)$ and $U_{A\to B}^\theta(x, t_f) = U_B(x)$. We propose to parameterise both components by learning $b_t^\phi$ using standard Conditional Flow Matching (CFM) (Lipman et al., 2023) and $U_t^\theta$ using an extension of CFM, which we call Conditional Density Matching.

### 3.1 LEARNING $b_t^\theta$ USING CONDITIONAL FLOW MATCHING

Conditional Flow Matching (CFM) is a general framework for learning a vector field $v_t^\phi$ that drives samples from one arbitrary distribution $p_0$ to another $p_1$ along a set of time-dependent intermediate distributions $p_t$. As it can generally be assumed that there is no access to the ground truth vector field $v_t$ or samples from the intermediate distributions $p_t$ beyond the initial and final distributions, CFM approaches this by considering a *conditional* time-dependent distribution $p_t(x_t \mid z)$ generated by a *conditional* vector field $v_t(x_t \mid z)$ and a coupling distribution $q(z)$. A common choice is to have $q(z)$ defined as an Optimal Transport coupling (Tong et al., 2024) to enforce $v_t$ to follow straight paths.

Using this coupling of conditional and marginal vector fields, Lipman et al. (2023) showed that if we learn the vector field $v^\phi(x_t, t)$ by regressing on the conditional vector field $v_t(x_t, t \mid z)$ using

$$\mathcal{L}_{\text{CFM}} = \mathbb{E}_{t \sim \text{Uni}(0,1),\, z \sim q(z),\, x_t \sim p_t(x_t \mid z)} \left[ \left\| v^\phi(x_t, t) - v_t(x_t \mid z) \right\|^2 \right] \tag{14}$$

then this is equivalent to regressing on the vector field $v_t(x_t)$ directly.

Crucially, given a paired conditional vector field and conditional distribution, under minor regularity conditions, for any choice of coupling distribution $q(z)$ the marginal vector field $v_t(x_t) = \left\langle \frac{v_t(x_t|z)p_t(x_t|z)}{p_t(x_t)} \right\rangle_{q(z)}$ and marginal distribution $p_t(x_t) = \langle p_t(x_t|z) \rangle_{q(z)}$ are shown to jointly solve the continuity equation (Tong et al., 2024). As such, if we set $p_0 = p_A$, $p_1 = p_B$, $t_f^{-1} v^\phi(x, s) = b^\phi(x, t)$ under the rescaling $t = s t_f$, and define $p_s(x) = p(x, t) \propto \exp\left(-\beta U^\theta_{A \to B}(x, t)\right)$, then CFM provides a valid approach to finding the escorting vector field $b^\phi$ for our escorted switching protocol.

## 3.2 Learning $U^\theta_t$ using Conditional Density Matching

This still leaves us with the problem of learning the time-dependent potential $U^\theta(x, t)$. For this purpose, we employ a similar trick to that used by CFM to learn the time-dependent potential $p^\theta(x, t) \propto \exp\left(-\beta U^\theta_{A \to B}(x, t)\right)$ from the conditional time-dependent distribution $p_t(x_t \mid z)$ and consider the following extended version of the Maximum Likelihood Estimation (MLE) objective

$$\mathcal{L}_{\text{DM}} = \mathbb{E}_{t \sim \text{Uni}(0,1),\, x_t \sim p_t(x_t)} \left[ -\log p^\theta(x_t, t) \right], \tag{15}$$

which adds an additional time dependence. To highlight the similarity with Flow Matching, we denote this object as the **Density Matching (DM)** objective.

Due to $\mathcal{L}_{\text{DM}}$ being defined as an expectation over the unknown distribution $p_t(x_t)$, learning $U^\theta_{A \to B}(x, t)$ directly using this objective is not possible. However, similar to the CFM objective, we can equivalently express another maximum likelihood objective using the conditional distribution $p_t(x_t \mid z)$ instead. We denote this as the **Conditional Density Matching (CDM)** objective:

$$\mathcal{L}_{\text{CDM}} = \mathbb{E}_{t \sim \text{Uni}(0,1),\, z \sim q(z),\, x_t \sim p_t(x_t|z)} \left[ -\log p^\theta(x_t, t) \right]. \tag{16}$$

Similar to the CFM objectives, the DM and CDM objectives have equivalent gradients, $\nabla_\theta \mathcal{L}_{\text{DM}} = \nabla_\theta \mathcal{L}_{\text{CDM}}$, and thus MLE using the conditional distribution will result in the same learned marginal distribution $p^\theta_t$ as MLE using the marginal distribution.

In summary, combining the CFM objective reviewed above with the proposed CDM objective, we thus learn a pair of escorting vector field $b^\phi(x, t)$ and potential $U^\theta_{A \to B}(x, t)$ that collective solve the continuity equation. When these two components are collectively used as the escorted switching protocol in an E-NEQ estimator, this results in a low-variance estimate of $\Delta F_{A \to B}$.

## 4 Efficient Multi-State Free-Energy Estimation

With the previous section proposing a method for learning the escorted protocol $(U^\theta_{A \to B}, b^\phi)$, we now consider some practical restrictions of E-NEQ estimators. Namely, (i) the work calculation becoming prohibitively expensive because a very small global time step $dt$ being required to deal with the numerical instability of the stationary distribution preserving dynamics, and (ii) in the multi-state setting the number of escorted switching protocols grows exponentially with the number of states. We will address both of these issues in the following two sections. In the appendix we further discuss three more standard, but important, practical considerations.

### 4.1 Efficient Work Calculation by Lie-Trotter Splitting

Molecular Dynamics simulation is known to quickly become numerically unstable when using large time-steps due to the potential energy including sharply peaked components such as Lennard–Jones potentials. As such, when simulating the dynamics of our escorted protocol in Eq. (9), we are required to use a very small global time-step $dt$. Unfortunately, as a result of this, in the work calculation we therefore must evaluate the divergence many times, which can be computationally very expensive for any system of considerable size. Ideally, we would therefore like to decouple the divergence calculation from the unstable stationary distribution preserving dynamics.

For this purpose we propose to use Lie–Trotter (Trotter, 1959) splitting to split the combined dynamics into two separate steps; first a step using $b(x, t)$ and then a step using the stationarity-preserving distribution preserving dynamics. Given our escorted dynamics $\widehat{\mathcal{L}}_t = \mathcal{L}_t + \mathcal{E}_t$ where

$\mathcal{E}_t = -\nabla \cdot (b(x,t)\rho(x,t))$ is the escort transport, a Lie–Trotter step of size $h = \mathrm{d}t$ frozen at time $t$ approximates the full evolution by composing the subflows as

$$e^{h\widehat{\mathcal{L}}_t} \approx e^{h\mathcal{L}_t} e^{h\mathcal{E}_t}. \tag{17}$$

Under such split dynamics, a simplified version of the Escorted Jarzynski Equality holds.

---

**Theorem 4.1** (Split Escorted Jarzynski Equality). *Given an escorted switching protocol $(U_{A\to B}, b)$ and split escorted time-dependent dynamics $e^{h\widehat{\mathcal{L}}_t} \approx e^{h\mathcal{L}_t} e^{h\mathcal{E}_t}$, where $\mathcal{L}_t$ is the stationarity-preserving operator (i.e. $\mathcal{L}_t p(x,t) = 0$ with $p(x,t) \propto e^{-\beta U_{A\to B}(x,t)}$) and $\mathcal{E}_t \rho = -\nabla \cdot (b(x,t)\rho)$ is the escort transport, consider a Lie–Trotter time grid $t_k = kh$ with $N \cdot h = t_f$ and the split update*

$$x'_k = x_k + b(x_k, t_k)h, \qquad x_{k+1} \sim K_{t_k}(x'_k, \cdot), \tag{18}$$

*where $K_t$ is any transition kernel that preserves $p(x,t)$ at frozen time $t$. If $x_0 \sim p_A(x)$, then*

$$\left\langle e^{-\beta \widehat{W}_{A\to B}(\mathbf{x}, h)} \right\rangle_{(U_{A\to B}, b, h)} = e^{-\beta \Delta F_{A\to B}} + \mathcal{O}(h), \tag{19}$$

*where the split escorted work is defined by*

$$\widehat{W}_{A\to B}(\mathbf{x}, h) = \sum_{k=0}^{N-1} \left( \frac{\partial U_{A\to B}(x'_k, t_k)}{\partial t} - \beta^{-1}\nabla \cdot b(x_k, t_k) \right) h. \tag{20}$$

*In particular, (19) holds exactly in the limit $h \to 0$, recovering the Escorted Jarzynski Equality.*

---

*Proof.* Consider the escort step as a deterministic kernel $L_{t_k}(x'|x) = \delta_{x+b(x,t_k)h}(x')$ and the stationarity-preserving step as a kernel $K_{t_k}$ with stationary distribution $p(x, t_k)$ such that the composed update is the kernel $P_{t_k} = K_{t_k} \circ L_{t_k}$. Then the proof follows from the discrete-time E-JE in Vaikuntanathan and Jarzynski (2011). $\square$

While this still requires us to use the global time discretization $h = \mathrm{d}t$ for each individual substep as well as for the calculation of the work, it allows for smaller internal time discretization within the stationarity-preserving kernel $K_t$ independent of the escort step. This split can be used to significantly reduce the number of divergence evaluations $\nabla_x \cdot b(x,t)$ needed in the work calculation.

For reference, operator splitting is a common topic in MD simulation (Frenkel and Smit). It is, for example, at the core of the BAOAB splitting for underdamped Langevin dynamics and the velocity Verlet integrator for Hamiltonian Monte Carlo (Leimkuhler and Matthews, 2013; Swope et al., 1982).

## 4.2 EFFICIENT MULTI-STATE FE ESTIMATION USING FLOW-GRAPH BASED MBAR

With the combined CFM and CDM approach to learn the escorted switching protocol $(U^\theta_{A\to B}, b^\phi)$ between the two thermodynamic states $A$ and $B$, we now consider how to extend this to the case of estimating the relative free-energy difference $\{\Delta F_{i\to j}\}_{i=1, j=1}^K$ between a collection of $K$ thermodynamic states. Within the context of the drug-discovery pipeline, lead optimization for example often involves large assays of ligands for which we need to assess their binding affinity (Wang et al., 2015).

Naively, using a single reference state $k$ we can train $K-1$ escorted switching protocols $\{(U^{\theta_i}_{k\to i}, b^{\phi_i}_{k\to i})\}_{i=1}^K$ to calculate $\{\Delta\mathcal{F}_{i\to j}\}_{i=1, j=1}^K$ using the E-JE. Considering that Free-Energy differences are a state function, we can then obtain all not-directly connected estimates as $\Delta\mathcal{F}_{i\to j} = -\Delta\mathcal{F}_{k\to i} + \Delta\mathcal{F}_{k\to j}$. While this would result in a consistent estimate for all pairs of states, it is generally understood that to obtain accurate multi-state free-energy it is best to obtain all $\{W_{i\to j}\}_{i=1, j=1}^K$ individually and use the self-consistent Multistate Bennett Acceptance Ratio (MBAR) (Shirts and Chodera, 2008) to obtain $\{\Delta\mathcal{F}_{i\to j}\}_{i=1, j=1}^K$. Following this line of thinking, a straightforward extension of the CFM/CDM framework would therefore be to learn a $U^{\theta_{i,j}}_{i\to j}$ and escorting vector fields $b^{\phi_{i,j}}_{i\to j}$ for each pair of states $i$ and $j$ individually and use them to obtain $\hat{W}_{i\to j}$ according to Eq. (4). However, for a set of $K$ thermodynamic states, this would require $K(K-1)/2$ models, which would quickly become infeasible.

Instead, we propose to construct an Escorted Protocol Flow Graph.

**Definition 4.2** (Escorted Protocol Flow Graph (EPFG)). Given a collection of $K$ thermodynamic states and $K-1$ escorted switching protocols $\{(U_{k\to i}^{\theta_i}, b_{k\to i}^{\phi_i}\}_{i=1, i\neq k}^{K}$ with central node $k$ an **Escorted Protocol Flow Graph** (EPFG) is constructed by considering the $K$ thermodynamic states to be the nodes and the escorted switching protocols $\{(U_{k\to i}, b_{k\to i}\}_{i=1, j=1}^{K}$ between all states $i$ and $j$ to represent edges. Here all edges not connected to the central node $k$ are given by

$$(U_{i\to j}(x,t), b_{i\to j}(x,t)) = \begin{cases} (U_{k\to i}^{\theta_i}(x, t_f - 2t), -2b_{k\to i}^{\phi_i}(x, t_f - 2t)) & \text{if } 0 \leq t < \frac{t_f}{2} \\ (U_{k\to j}^{\theta_j}(x, 2t - t_f), 2b_{k\to j}^{\phi_j}(x, 2t - t_f)), & \text{if } \frac{t_f}{2} \leq t \leq t_f. \end{cases} \quad (21)$$

Using the EPFG and the escorted protocols defined as its edges, including the constructed concatenated ones, we can now obtain $\{W_{i\to j}\}_{i=1, j=1}^{K}$ for all combinations of states $i$ and $j$ while only requiring the training of $K-1$ protocols. This enables more accurate free-energy estimates using the MBAR estimator compared to using the pair based approach while minimizing the training costs.

## 5 EXPERIMENTS

We evaluate the feasibility of learning the thermodynamic flow network of coupled $U^\theta$ and escorting field $b^\phi$ using the well-known alanine dipeptide (ADP) system. While limited in size, ADP shows similar challenges to larger systems such as high-energy barriers and solvent effects. Furthermore, as ADP in solvent has six distinct metastable states $\alpha_L$, $\alpha_D$, $\beta$, $C5$ and $\alpha'$ as visualized in Fig. 2. These states are known to exhibit significant differences in conformational free energy it serves as an well-understood benchmark for the multi-state setting.

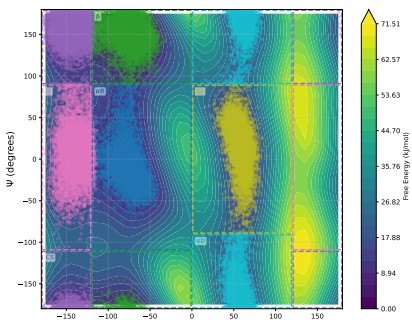

Figure 2: Umbrella sampling FES reconstruction and training samples for ADP.

**Data Generation** For each metastable state, we generated 10,000 samples using a harmonic flat-bottom constraint on the torsion angles defined by the metastable state boundaries, as specified in the appendix. Training samples for each state are visualized in Fig. 2. We observe that the training samples cover each state but do not represent the correct ratios as observed in full equilibrium sampling.

**Baselines** To obtain a baseline value to validate our method against, we performed an extensive Umbrella Sampling (US) estimate using a large number of windows and the same setup as used for data generation. Additionally, we compare our method against a neural version of Targeted Free-Energy Perturbation (TFEP). For our comparison we will use the vector field $b^\phi$ learned using Conditional Flow Matching to implement the TFEP method. As such, the only difference between TFEP and E-NEQ can come from the inclusion of the stationary distribution preserving dynamics.

**Model Details** We used the same model definitions and training setup for all learned escorted protocols. The escorting vector field $b^\phi$ is implemented using the SE(3)-equivariant graph neural network (Satorras et al., 2021) with an additional learnable time-embedding component to make it time-dependent (Tan et al., 2025). We use optimal transport coupling $q$ (Tong et al., 2024) to enforce that the escorting vector field and the time-dependent density not only collectively solve the continuity equation but also align with the dynamical optimal transport problem (Benamou and Brenier, 2000). While not discussed in depth in this work, there is a close connection between the dynamical optimal transport problem and the amount of dissipated work in escorted switching protocols (Zhong et al., 2023; Zhong and DeWeese, 2024). We leave exploring this interplay further for future work.

The potential $U^\theta$ is parameterised as the negative log probability of a discrete normalizing flow with conditional affine coupling layers and a similar learnable time-embedding component. We chose to use a discrete-time normalising flow here, instead of a more general energy-based model, to minimize the complexity of maximum-likelihood training.

**Integration Details** For the implementation of the escorted dynamics in Eq. (9), we use the following setup. The escorting vector field is implemented in all experiments using a Runge–Kutta integrator, while we consider three different options for the stationary preserving component of the split dynamics: underdamped Langevin, overdamped Langevin, and Hamiltonian Monte Carlo, all of which use Metropolis-Hastings (MH) correction to preserve the stationary distribution in finite time. The split operators $\mathcal{E}$ and $\mathcal{L}$ are iteratively applied for 100 time steps, during which the work is calculated.

Table 1: Quantitative results of the estimated $\Delta F$ between the central $\alpha_R$ state of the escorted flow graph and the directly connected states, as well as the Mean Absolute Error (MAE) over all pairs of states including those not directly connected. The $\mathcal{L}$ column denotes the type of integrator used: Underdamped Langevin (UL), Overdamped Langevin (OL) or Hamiltonian Monte Carlo (HMC). EPFG denotes whether the estimates were obtained using pairwise summation (EPFG=N), or using the escorted protocol flow graph (EPFG=Y).

| Method | $\mathcal{L}$ | EPFG | $\alpha_L$ | $\alpha_D$ | $\beta$ | $C5$ | $\alpha'$ | MAE |
|---|---|---|---|---|---|---|---|---|
| US | - | - | 7.42 ± 0.16 | 12.07 ± 0.40 | -1.11 ± 0.03 | 1.37 ± 0.05 | 6.55 ± 0.13 | - |
| TFEP | - | N | 8.60 ± 0.05 | 12.39 ± 0.06 | 0.77 ± 0.04 | 2.39 ± 0.04 | 5.78 ± 0.03 | 1.17 |
| E-NEQ | UL | N | 7.93 ± 0.13 | 12.82 ± 0.16 | 0.54 ± 0.12 | 1.47 ± 0.21 | 5.96 ± 0.06 | 0.93 |
| (ours) | UL | Y | **7.35 ± 0.12** | 12.54 ± 0.15 | 0.78 ± 0.11 | **1.31 ± 0.20** | **6.14 ± 0.06** | 0.88 |
| | OL | N | 8.22 ± 0.07 | 12.86 ± 0.08 | **0.26 ± 0.12** | 1.78 ± 0.10 | 5.59 ± 0.05 | 0.96 |
| | OL | Y | 8.06 ± 0.05 | **12.09 ± 0.06** | **0.24 ± 0.05** | 1.69 ± 0.05 | 7.04 ± 0.04 | **0.59** |
| | HMC | N | 7.96 ± 0.12 | 13.05 ± 0.17 | 0.62 ± 0.09 | **1.57 ± 0.22** | 5.97 ± 0.05 | 0.99 |
| | HMC | Y | 7.33 ± 0.11 | 12.89 ± 0.16 | 0.50 ± 0.09 | 1.83 ± 0.20 | 5.94 ± 0.06 | 0.95 |

## 5.1 RESULTS

In Tab. 1, we report the main quantitative results of learning the escorting vector field $b^\phi$ and corresponding potential $U^\theta$ for use in the E-NEQ estimator, compared to the TFEP approach. We note that the $\alpha_R$ state was chosen as the central node when constructing the flow graph. As such, all reported free-energy differences are relative to $\alpha_R$.

**E-NEQ improves over TFEP for the same amount of divergence calculations** Across all three integration approaches, we find that E-NEQ with learned escorted dynamics outperforms TFEP. This holds for both the directly reported $\Delta F$ values using the central $\alpha_R$ state and the MAE of the free-energy differences across all pairs of states, with the $\beta$ state a noticeable exception. In almost all cases the found free-energy differences closely resemble the US baseline with $\beta$ as notable exception.

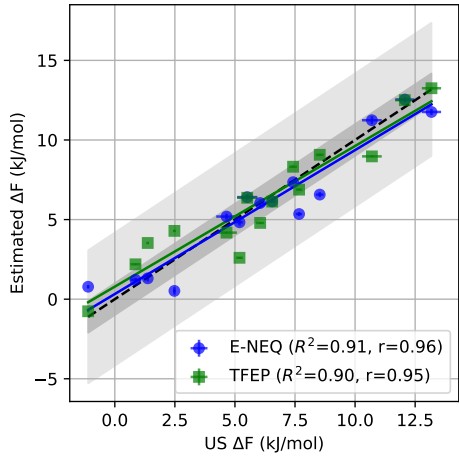

Figure 3: Correlation of estimated free-energy differences from E-NEQ and TFEP against the Umbrella Sampling MBAR baseline.

A further study of the correlation between the estimated free-energy differences and the true (as reported by US) free-energy differences similarly shows that the E-NEQ estimator outperforms TFEP. As highlighted in Fig. 3, both E-NEQ and TFEP show strong correlations, with $R^2$ and Pearson correlation coefficients of 0.91 and 0.96 for E-NEQ and 0.90 and 0.95 for TFEP. Both methods are firmly within 1 kcal/mol of the true free-energy differences, and in most cases even within 1 kJ/mol. This is well within the generally considered acceptable tolerances for free-energy estimation.

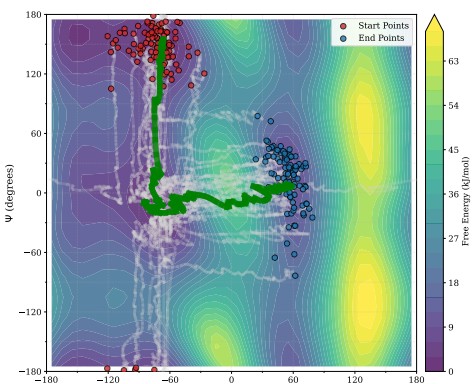

Figure 4: Trajectories of the E-NEQ estimator using a concatenated protocol starting from the $\beta$ state towards the $\alpha_L$ state using the underdamped Langevin integrator.

**Concatenating Protocols improves Multistate Free-Energy Estimation** Comparing the results in table 1 between using pairwise consistent estimation between all states using only directly connected states (EPFG=N) and using a EPFG with concatenated protocols (EPFG=Y), we find that the latter is in almost all cases more accurate. Notably, it is not only the MAE that improves, but also the estimated free-energy differences between the states directly connected to the reference $\alpha_R$ state, despite using the same trajectories.

In Fig. 4 we visualise E-NEQ trajectories starting from the $\beta$ state towards the $\alpha_L$ state following a concatenated protocol. The trajectories show the concatenation of the characteristically straight paths of OT-CFM starting from the $\beta$ and visiting the $\alpha_R$ state before finally transitioning to the $\alpha_L$ state.

Table 2: Number of successful trajectories starting in each state for the different integration schemes. The maximum number of trajectories is 50,000, 10,000 to each other state.

| $f_t^U$ | dt | acc. rate | $\alpha_R$ | $\alpha_L$ | $\alpha_D$ | $\beta$ | $C5$ | $\alpha'$ |
|---|---|---|---|---|---|---|---|---|
| Underdamped Langevin | 1e-4 | 0.94 | 49459 | 48871 | 48383 | 49183 | 48328 | 49419 |
| Overdamped Langevin | 1e-8 | 0.91 | 49535 | 49420 | 49265 | 49414 | 49172 | 49539 |
| Hamiltonian Monte Carlo | 1e-4 | 0.96 | 44056 | 46892 | 38892 | 40145 | 32477 | 47582 |

**Stationary Preserving Dynamics comparison** Lastly, we observe that, within the E-NEQ estimator with learned dynamics, underdamped and overdamped Langevin seem to be more stable than Hamiltonian Monte Carlo. Specifically, we found that over longer transitions, as occurs during the concatenation of protocols for multistate free-energy estimation, the HMC approach was more likely to diverge, resulting in discarded samples and ultimately a higher finite sample bias. The discrepancy between the number of successful trajectories starting in each state between the different integrators is reported in Tab. 2. MH correction does not help in this case due to the instability primarily caused by the deterministic vector field putting the samples in high-energy regions of the learned potential. As exemplified by the extremely small step size for overdamped Langevin dynamics, our proposed Lie-Trotter splitting of the work calculation is a necessary step to make the computation feasible within an acceptable compute budget.

## 6 DISCUSSION

In this work we have proposed a method to learn the escorting switching protocol $(U_{A \to B}^\theta, b^\phi)$ to construct an E-NEQ estimator with minimal variance. For this purpose we considered the framework of Conditional Flow Matching to learn $b^\phi$ and proposed an extension named Conditional Density Matching to learn $U_{A \to B}^\theta$. Furthermore, we considered two practical considerations in the form of Lie-Trotter splitting to reduce the computational cost of the work calculation and a flow graph construction to reduce the combinatorial complexity of learning multiple protocols in the multi-state setting. In our experimental evaluation using the ADP system, which has six well-defined metastable states, we have shown our proposed method for learning the escorted protocols to be effective.

**Limitations and future work** While ADP is a fitting benchmark for our method due to its multiple metastable states, similar challenges to larger systems, and generally a complexity level similar in size to studied in other related work (Rizzi et al., 2021; 2023; Máté et al., 2024; Erdogan et al., 2024; He et al., 2025), it is still a toy problem within the context of practical applications. We therefore emphasize the need to focus on scalability in future work. Preliminary experiments have shown that to achieve this emphasis should be placed on accurately learning the correct time-dependent potential $U_{A \to B}^\theta$. For this purpose we believe that recent advances in scaling discrete time normalising flows is an important avenue to explore (Rehman et al., 2025; Zhai et al., 2024).

In addition to this, we would like to note that in this work we choose to align the proposed method as closely as possible to the presented theory in Sec. 2 to provide a well grounded first exploration of neural escorted free-energy estimation. As such, we believe that by lifting some of these strict constraints performance could be improved. For example, removing the Metropolis-Hasting step that is used to ensure that stationary is preserved under discretized dynamics, and using adaptive ODE solvers instead of Runga-Kutta to set the lie-trotter splitting could therefore be beneficial. Both of these are common practice for standard molecular dynamic simulation. Furthermore, extensions of the current framework through hard constraints for enforcing the continuity equation instead of just encoding it in the joint CFM and CDM learning objectives, or incorporating the EPFG already during the training process could be beneficial. Lastly, we note that while we consider Flow Matching as the base for our E-NEQ estimation other generative modelling approaches in the form of Schrodinger Bridge marching and recent advances in Flow-Maps should also be considered in future work.

### ACKNOWLEDGMENTS

LH is supported by the EPSRC Centre for Doctoral Training in Autonomous Intelligent Machines and Systems (EP/S024050/1). MB and LH are partially supported by the EPSRC Turing AI World-Leading Research Fellowship No. EP/X040062/1 and EPSRC AI Hub No. EP/Y028872/1. NMA is supported by an MRC Enterprise (Industrial CASE) Studentship (MSD2021_1538327) at the University of Oxford.

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

## A APPENDIX

### A.1 BENNETT ACCEPTANCE RATIO (BAR) AND MULTISTATE BENNETT ACCEPTANCE RATIO (MBAR)

In this section we briefly introduce the Bennett Acceptance Ratio (BAR) and Multistate Bennett Acceptance Ratio (MBAR) estimators as introduced in the main text. First, the BAR estimator can be interpreted as a maximum likelihood solution for the logistic model that is implied by the Crooks Fluctuation Theorem (CFT). Given a collection of forward $\mathbf{x}_i^F$ and backward $\mathbf{x}_i^B$ trajectories and their corresponding work values $W_{A\to B}(\mathbf{x}_i^F)$ and $W_{B\to A}(\mathbf{x}_i^B)$ respectively, BAR solves:

$$\sum_{i=1}^{N_A} \frac{1}{1 + \exp\left(\beta\left(W_{A\to B}\left(\mathbf{x}_i^F\right) - \Delta F\right) + c\right)} = \sum_{i=1}^{N_B} \frac{1}{1 + \exp\left(\beta\left(-W_{B\to A}\left(\mathbf{x}_i^B\right) - \Delta F\right) - c\right)} \tag{22}$$

where $c = \ln\frac{N_A}{N_B}$ and is equal to 0 for all our experiments. To find $\Delta F$ we can now use standard root-finding algorithms. The approximate solution for $\Delta F$ found using this estimator is known to have minimal variance among estimators that use both forward and backward trajectories (Bennett, 1976).

Adjusted to the multistate setting, the MBAR estimate is given by the set of equations (Shirts and Chodera, 2008):

$$F_k = -\beta^{-1}\ln\left(\sum_{j=1}^{K}\sum_{n=1}^{N_j} \frac{\exp\left(-\beta W_{i\to j}(\mathbf{x}_n^{i\to j})\right)}{\sum_{l=1}^{K} N_l \exp\left(F_l - \beta W_{l\to j}(\mathbf{x}_n^{l\to j})\right)}\right) \tag{23}$$

where $W_{i\to j}(\mathbf{x}_n^{i\to j})$ is the work done on the $n$-th trajectory moving from state $i$ to state $j$. Note that here we obtain absolute free-energies as $F_l$ with respect to the reference state $k$. Thus, to obtain the free-energy differences between all pairs of states, we can use the following equation:

$$\Delta F_{i\to j} = F_j - F_i. \tag{24}$$

### A.2 VALIDITY OF METHOD

In Sec. 3 we showed an approach to learn the potential $U^\theta$ and vector field $b^\phi$ that can collectively be used as an escorted protocol for free-energy estimation. For this purpose we aligned the objective of learning the vector field with standard Conditional Flow Matching, and introduced an extension

named Conditional Density Matching to learn the potential / density. In this section we provide a more formal derivation to show that this forms a valid method.

For this purpose we first give a formal derivation of Conditional Density Matching, and subsequently show how jointly using CFM and CDM provides a valid escorted protocol.

### A.2.1 CONDITIONAL DENSITY MATCHING

With $q(z)$ the coupling distribution and $p_t(x_t \mid z)$ the conditional probability path generated by a conditional vector field $v_t(x_t \mid z)$, the marginal probability path is defined as follows:

$$p_t(x_t) = \int p_t(x_t \mid z) \, q(z) \, \mathrm{d}z. \tag{25}$$

Our objective is to parameterize a density $p_t^\theta$ (or equivalently, a potential $U_t^\theta$ via $p_t^\theta \propto \exp(-\beta U_t^\theta)$) and train it to approximate $p_t$ using Maximum Likelihood Estimation. We call the approach to directly do this **Density Matching**,

$$\mathcal{L}_{\mathrm{DM}} = \mathbb{E}_{t \sim \mathrm{Uni}(0,1), \, x_t \sim p_t(x_t)} \left[ -\log p^\theta(x_t, t) \right], \tag{26}$$

but as discussed in the main body of the paper, this is infeasible as we are not able to directly sample from $p_t$. This is where our alternative learning method, **Conditional Density Matching** (CDM), comes in. CDM is defined using the conditional probability path and the coupling distribution, which we have direct access to:

$$\mathcal{L}_{\mathrm{CDM}} = \mathbb{E}_{t \sim \mathrm{Uni}(0,1), \, z \sim q(z), \, x_t \sim p_t(x_t|z)} \left[ -\log p^\theta(x_t, t) \right]. \tag{27}$$

Crucially, as mentioned in the main paper but not further elaborated on, the DM and CDM objectives have equivalent gradients, $\nabla_\theta \mathcal{L}_{\mathrm{DM}} = \nabla_\theta \mathcal{L}_{\mathrm{CDM}}$. Similar to the argument used for CFM, we can thus use the CDM loss for optimization and consequently also optimize for the DM loss.

We now formally show that these gradients are the same.

**Proposition A.1** (CDM Gradient Equivalence)**.** *The Density Matching and Conditional Density Matching objectives,*

$$\mathcal{L}_{\mathrm{DM}} = \mathbb{E}_{t \sim \mathrm{Uni}(0,1), \, x_t \sim p_t(x_t)} \left[ -\log p^\theta(x_t, t) \right], \tag{28}$$

$$\mathcal{L}_{\mathrm{CDM}} = \mathbb{E}_{t \sim \mathrm{Uni}(0,1), \, z \sim q(z), \, x_t \sim p_t(x_t|z)} \left[ -\log p^\theta(x_t, t) \right], \tag{29}$$

*have identical gradients with respect to $\theta$:*

$$\nabla_\theta \mathcal{L}_{\mathrm{DM}} = \nabla_\theta \mathcal{L}_{\mathrm{CDM}}. \tag{30}$$

*Proof.* We show the stronger statement that $\mathcal{L}_{\mathrm{DM}} = \mathcal{L}_{\mathrm{CDM}}$. Starting from $\mathcal{L}_{\mathrm{CDM}}$:

$$\mathcal{L}_{\mathrm{CDM}} = \mathbb{E}_{t \sim \mathrm{Uni}(0,1), \, z \sim q(z), \, x_t \sim p_t(x_t|z)} \left[ -\log p^\theta(x_t, t) \right] \tag{31}$$

$$= \mathbb{E}_{t \sim \mathrm{Uni}(0,1)} \left[ \iint p_t(x_t \mid z) \, q(z) \left( -\log p^\theta(x_t, t) \right) \, \mathrm{d}z \, \mathrm{d}x_t \right] \tag{32}$$

$$= \mathbb{E}_{t \sim \mathrm{Uni}(0,1)} \left[ \int \left( \int p_t(x_t \mid z) \, q(z) \, \mathrm{d}z \right) \left( -\log p^\theta(x_t, t) \right) \, \mathrm{d}x_t \right] \tag{33}$$

$$= \mathbb{E}_{t \sim \mathrm{Uni}(0,1)} \left[ \int p_t(x_t) \left( -\log p^\theta(x_t, t) \right) \, \mathrm{d}x_t \right] \tag{34}$$

$$= \mathbb{E}_{t \sim \mathrm{Uni}(0,1), \, x_t \sim p_t(x_t)} \left[ -\log p^\theta(x_t, t) \right] \tag{35}$$

$$= \mathcal{L}_{\mathrm{DM}}. \tag{36}$$

Following this, we now directly have that $\nabla_\theta \mathcal{L}_{\mathrm{DM}} = \nabla_\theta \mathcal{L}_{\mathrm{CDM}}$. $\square$

What is crucial here is that we can do this at the same time as the learning of the velocity field $b^\phi$. At every epoch, we sample $(x_0, x_{t_f}) = z \sim q(z)$, $t \sim \mathrm{Uni}(0, 1)$, and consequently use our definition of the conditional probability path to obtain $x_t \sim p_t(x_t \mid z)$, which can now be used to learn both $b^\phi$ using the CFM loss and $p^\theta$ using the CDM loss.

### A.2.2 JOINTLY USING CFM AND CDM PROVIDES A VALID ESCORTED PROTOCOL

We now focus on how CFM and CDM can be used to jointly train the escorting vector field $b^\phi$ via CFM and the time-dependent potential $U^\theta$ via CDM to produce a pair $(U^\theta_{A\to B}, b^\phi)$ that provides a valid escorted protocol as defined in Thm. 2.3. We first recall the key criteria for such a protocol:

1. **Boundary conditions on $U$:** $U_{A\to B}(x,0) = U_A(x)$ and $U_{A\to B}(x, t_f) = U_B(x)$.
2. **Boundary conditions on $b$:** $b(x,0) = b(x, t_f) = 0$.

Notably, if these two boundary conditions are satisfied, any choice of $(U_{A\to B}, b)$ works as a valid escorted switching protocol. A hard requirement on how the two are related only comes in when we want to achieve zero variance in the estimator, as discussed in Thm. 2.5. In this case we add the following requirement:

3. **Continuity equation:** $\partial_t p(x,t) + \nabla \cdot (p(x,t)\, b(x,t)) = 0$, where $p(x,t) \propto \exp(-\beta U_{A\to B}(x,t))$.

As the two boundary conditions can trivially be satisfied, we now show that the CFM and CDM learning frameworks collectively target the continuity condition. For this purpose we first restate a well-known result from Conditional Flow Matching:

---

**Theorem A.2** (Conditional continuity equation implies marginal continuity equation). *Let $q(z)$ be a coupling distribution over pairs $z = (x_0, x_1)$ with $x_0 \sim p_A$ and $x_1 \sim p_B$, and let the pair $\{p_t(x_t \mid z), v_t(x_t \mid z)\}$ be a conditional probability path and conditional vector field such that for each $z$ the conditional continuity equation*

$$\partial_t p_t(x_t \mid z) + \nabla \cdot \big(p_t(x_t \mid z)\, v_t(x_t \mid z)\big) = 0 \tag{37}$$

*is satisfied. Then with the marginal probability path and marginal vector field defined as*

$$p_t(x_t) = \langle p_t(x_t \mid z) \rangle_{q(z)}, \tag{38}$$

$$v_t(x_t) = \left\langle \frac{v_t(x_t \mid z)\, p_t(x_t \mid z)}{p_t(x_t)} \right\rangle_{q(z)}, \tag{39}$$

*the pair $(p_t(x_t), v_t(x_t))$ collectively solves the marginal continuity equation*

$$\partial_t p_t(x_t) + \nabla \cdot \big(p_t(x_t)\, v_t(x_t)\big) = 0. \tag{40}$$

*Proof.* See, for example, Tong et al. (2024). $\qquad\square$

---

Note here that it is crucial that the conditional probability path $p_t(x_t \mid z)$ and conditional vector field $v_t(x_t \mid z)$ are chosen consistently, i.e., such that they jointly satisfy the conditional continuity equation (37). A standard choice is the optimal transport (OT) conditional path with linear interpolation $x_t = (1-t)x_0 + t\, x_1$ for $z = (x_0, x_1)$, which gives

$$p_t(x_t \mid z) = \mathcal{N}(x_t \mid (1-t)x_0 + t\, x_1,\, \sigma^2), \qquad v_t(x_t \mid z) = x_1 - x_0, \tag{41}$$

where $\sigma > 0$ is a small fixed variance. This pair satisfies (37) by construction, as the conditional density is simply transported along the constant velocity field $v_t = x_1 - x_0$.

We can now state the validity of the joint CFM/CDM escorted protocol as a direct consequence of the above Theorem A.2 and Theorem A.1:

---

**Corollary A.3** (Validity of joint CFM/CDM escorted protocol). *Under the same conditions as the above theorem, let $v^{\phi^*}$ minimise $\mathcal{L}_{\mathrm{CFM}}$ and $p^{\theta^*}$ minimise $\mathcal{L}_{\mathrm{CDM}}$. Then, under the identification $b^\phi(x,t) = t_f^{-1} v^\phi(x, t/t_f)$ and $p^\theta(x,t) \propto \exp(-\beta U^\theta(x,t))$, the escorted protocol $(U^{\theta^*}, b^{\phi^*})$ satisfies the conditions of Theorem 2.5.*

---

*Proof.* By the standard CFM result, $v^{\phi^*}(x_t, t) = v_t(x_t)$ recovers the marginal vector field. By Thm. A.1 and the optimality of maximum likelihood estimation, $p^{\theta^*}(x_t, t) = p_t(x_t)$ recovers the marginal density. By the above theorem, this pair jointly solves the continuity equation $\partial_t p_t + \nabla \cdot (p_t\, v_t) = 0$. The time rescaling preserves the continuity equation, giving $\partial_t p^{\theta^*} + \nabla \cdot (p^{\theta^*}\, b^{\phi^*}) = 0$, which is exactly the condition of Theorem 2.5. $\qquad\square$

The above analysis shows that, with an appropriately chosen conditional probability path and conditional vector field that collectively solve the conditional continuity equation, we recover an optimal escorted switching protocol.

### A.3 FULL ALGORITHMIC OVERVIEW

Below we briefly provide the full algorithmic overview for multistate free-energy estimation using E-NEQ. Code is available at: https://github.com/LarsHoldijk/LEPFE

#### A.3.1 TRAINING

1. **Generating Training Samples.** Given a set of $K$ thermodynamic states of interest and their corresponding potential energies $\{U_k\}_{k=1}^{K}$, generate a dataset of $N_k$ equilibrium samples $\{\mathbf{x}_n^k\}_{n=1}^{N_k}$ from each state $k$ using standard molecular dynamics simulations.

2. **Learning $K-1$ Escorted Protocols.** Select one of the thermodynamic states as the reference state $k$ and learn the $K-1$ escorted protocols $\{(U_{k\to i}^{\theta_i}, b_{k\to i}^{\phi_i})\}_{i\neq k}$ using Conditional Flow Matching and Conditional Density Matching as proposed in Sec. 3.

3. **Constructing the Flow Graph.** Construct the fully connected Escorted Protocol Flow Graph by concatenating the learned protocols to define edges between all pairs of states, as defined in Eq. (21).

#### A.3.2 OBTAINING $\Delta F_{i\to j}$

1. **Running the Escorted Protocols.** For each pair of states $(i, j)$, run fixed-length non-equilibrium trajectories using the escorted dynamics in Eq. (9) (forward) and Eq. (13) (reverse), and compute the escorted work for each trajectory.

2. **Estimating Free-Energy Differences.** Use the collected forward and reverse work values $\{\hat{W}_{i\to j}\}_{i,j=1}^{K}$ with the MBAR estimator to obtain a self-consistent estimate of the free-energy differences between all pairs of states.

#### A.3.3 PRACTICAL DETAILS

So far we have assumed each thermodynamic state $k$ to have its own potential energy function $U_k$. In practice, this is only the case for a small number of free-energy estimation problems, such as solvation free-energy. For most other forms of free-energy, such as conformational free-energy which we consider in our experimental evaluation, the states have the same potential energy $U$ but correspond to different regions of the phase-space. In this case, a state-restricted potential energy function $U_i$ has to be defined as:

$$U_i(x) = \begin{cases} U(x), & x \in \Omega_i \\ +\infty, & x \notin \Omega_i \end{cases} \tag{42}$$

where $\Omega_i$ is the region of space that is considered to be part of thermodynamic state $i$. Restricting potential energies to specific regions of space is a common technique in Molecular Dynamics simulations (Torrie and Valleau, 1977).

**Equivariances** Molecular systems are in general invariant to rigid-body transformations, such as rotations and translations. Including these transformations in architecture design of $b^\theta$ and $U^\theta$ is thus desirable and can significantly improve the performance of the method. However, in the case of Flow Matching extra care needs to be taken to ensure that the target vector field $v_t$ also takes these transformations into account. This is formalised in the framework of Riemannian Flow Matching (Chen and Lipman, 2024; Bose et al., 2024).

**Thermodynamic Irrelevant Degrees of Freedom** In addition to the reduction in complexity that can be achieved by considering the symmetry structure of the system, an additional inductive bias can be introduced by considering the degrees of freedom of the system that do not contribute to the free-energy difference between the states of interest. For example, in the case of Alanine Dipeptide (ADP) a number of Carbon atoms are each connected to a single other heavy atom and have their remaining valences satisfied by hydrogen atoms. While these Hydrogen atoms considerably fluctuate during the simulation and can cause significant spikes in the potential energy, they generally have a

uniform contribution to the free-energy across all the states of interest. As such, we do not have to include them in the representation of the system used for $U^\theta$ and $b^\theta$.

Note that removing degrees of freedom that are considered thermodynamic irrelevant has to be done with care, as incorrect removal can have a detrimental effect on the performance of the method. For example, while the atoms in the ADP system attached to terminal Carbon atoms are considered thermodynamic irrelevant, the same is not true for the atoms attached to central heavy atoms.

## A.4 EXPERIMENTAL SETUP

### A.4.1 MOLECULAR DYNAMICS SIMULATION DETAILS

The simulation was run at 300 K using a standard Langevin middle integrator with a BAOAB splitting scheme and a time step of 2 fs. Samples were saved every 1000 steps. The Amber 14-alndx force-field was used with the GBn2 implicit solvent model. The same force field and implicit solvent model were used as the $U_A$ and $U_B$ potentials in the Umbrella Sampling simulations."

### A.4.2 METASTABLE STATE DEFINITIONS

For the Alanine Dipeptide (ADP) system, we use the following definition of the metastable states based on the $\phi$ and $\psi$ dihedral angles:

$$\alpha_R = \{-120 \leq \phi \leq 0, -110 \leq \psi \leq 90\} \tag{43}$$

$$\beta = \{-120 \leq \phi \leq 0, 90 \leq \psi \leq 180\} \cup \{-120 \leq \phi \leq 0, -180 \leq \psi \leq -110\} \tag{44}$$

$$C5 = \{-180 \leq \phi \leq -120, -180 \leq \psi \leq -110\} \cup \{-180 \leq \phi \leq -120, 90 \leq \psi \leq 180\}$$
$$\cup \{120 \leq \phi \leq 180, -180 \leq \psi \leq -110\} \cup \{120 \leq \phi \leq 180, 90 \leq \psi \leq 180\} \tag{45}$$

$$\alpha' = \{-180 \leq \phi \leq -120, -110 \leq \psi \leq 90\} \cup \{120 \leq \phi \leq 180, -110 \leq \psi \leq 90\} \tag{46}$$

$$\alpha_L = \{0 \leq \phi \leq 120, -90 \leq \psi \leq 90\} \tag{47}$$

$$\alpha_D = \{0 \leq \phi \leq 120, -180 \leq \psi \leq -90\} \cup \{0 \leq \phi \leq 120, 90 \leq \psi \leq 180\} \tag{48}$$

adapted from (Vymětal and Vondrášek, 2010).

We have visualised the 5 metastable states in figure 2.

## A.5 STANDARD JARZYNSKI EQUALITY RESULTS

As an additional baseline to compare our E-NEQ method against we consider using only the potential $U^\theta$ learned with the proposed density matching method as the switching protocol in the Jarzynski Equality. We only consider overdamped Langevin dynamics, and compare 3 different number of interpolation steps: 500, 5,000 and 50,000. The results are presented in Tab. 3.

Even at 50,000 steps, JE does not reach the accuracy of E-NEQ at 100 steps (MAE 0.93), highlighting the benefit of the learned escorting field.

Table 3: Extended Baseline quantitative results of the estimated $\Delta F$ between the central $\alpha_R$ state of the escorted flow graph and the directly connected states, as well as the Mean Absolute Error (MAE) over all pairs of states including those not directly connected.

| Method | steps | $\alpha_L$ | $\alpha_D$ | $\beta$ | $C5$ | $\alpha'$ | MAE |
|--------|-------|-----------|-----------|---------|------|-----------|-----|
| US | | $7.42 \pm 0.16$ | $12.07 \pm 0.40$ | $-1.11 \pm 0.03$ | $1.37 \pm 0.05$ | $6.55 \pm 0.13$ | - |
| TFEP | | $8.60 \pm 0.05$ | $12.39 \pm 0.06$ | $0.77 \pm 0.04$ | $2.39 \pm 0.04$ | $5.78 \pm 0.03$ | 1.17 |
| JE | 500 | $10.87 \pm 0.75$ | $1.62 \pm 0.57$ | $2.13 \pm 0.38$ | $-2.25 \pm 0.58$ | $4.25 \pm 0.10$ | 6.16 |
| JE | 5,000 | $-0.21 \pm 0.79$ | $16.83 \pm 0.94$ | $7.88 \pm 0.46$ | $-4.29 \pm 0.73$ | $4.95 \pm 0.09$ | 7.73 |
| JE | 50,000 | $3.27 \pm 0.61$ | $13.82 \pm 1.27$ | $3.07 \pm 1.44$ | $-1.97 \pm 1.30$ | $4.58 \pm 0.15$ | 3.92 |

