# OpenReview forum: "Learning Escorted Protocols For Multistate Free-Energy Estimation"
_ICLR.cc/2026/Conference — ICLR 2026 Poster_

### Official Review · Reviewer_X5Rb · 2025-10-26

**Soundness:** 3
**Presentation:** 4
**Contribution:** 3
**Rating:** 6
**Confidence:** 3

**Summary:**

A paper proposing an original solution to an important computational physics problem, namely computing free energy changes in non-equilibrium thermodynamics. The proposed solution uses flow matching techniques in an innovative way to suggest a more efficient/ accurate algorithm to sample Monte Carlo trajectories over which to estimate such free energy changes.

**Strengths:**

I really enjoyed reading this paper, albeit as a somewhat knowledgeable outsider. The authors make a good effort to provide a comprehensive background accessible to non-specialists (although I wonder how well it fares with non-physicists). The idea of using ML to minimise the Jaczinsky lower bound is elegant and the proposed improvements in terms of adopting larger time steps/ extending to multi-state estimation are potentially important for the community. The empirical validation is well carried out albeit not extensive (similar in that to physics papers)

**Weaknesses:**

The main weakness to me is how much this paper could appeal outside of the computational physics community, which is but a (small) strand in the ICLR community. Some opportunities to broaden the appeal of the paper are listed below in the questions.

**Questions:**

- Is it conceivable that your algorithmic improvements to learning the escorting protocol could be extended to other common ML scenarios where distributions are to be matched, e.g. diffusion models?
- Your approach is a practical solution to minimising the bias/ variance of the Monte-Carlo Jaczinsky estimator, does it come with guarantees/ special scenarios in which it could turn out to be exact?
- I didn't get much rationale for the choice of flow architecture etc, which potentially could be a factor in determining efficiency gains. Did you just take an off-the-shelf approach or are there mileage in optimising that side?

---

> ### Author Response · Authors · 2025-11-28
> **Initial response by authors**
>
> We thank the reviewer for their positive review of our paper. We appreciate the reviewer’s comments regarding the importance of the practical problem we are trying to address and the elegance of the proposed solution. Likewise, we are glad to hear that the reviewer found the paper enjoyable to read. As we discuss below, we deemed readability to be especially important to build understanding of the free-energy estimation problem in the ML and AI community, and are therefore excited to hear that the reviewer believes we have succeeded here.
>
> We would like to note that we understand the reviewer’s concerns regarding the appeal of the paper to the wider ICLR community at this time, and have addressed the reviewer’s list of questions relating to this below. We would, however, like to state that we personally believe that the limited interest in the problem of free-energy estimation so far within the ML community should be a reason in favor of acceptance, and not against. As the reviewer seems to agree, free-energy estimation is an incredibly important subject within the computational chemistry community and, as we show in this work, is closely related to problems that are broadly considered in the machine learning community. We believe our work can help build a line of research that will help bring new solutions to the free-energy estimation problem. Similar to how we have seen new advances being brought to the field of protein folding [1], docking [2], and transition path sampling [3, 4], which also initially were considered to be outside of the scope of mainline machine learning conferences such as NeurIPS, ICLR, and ICML.
>
> **Regarding the conceivability of our improvements to be connected to other works:**
>
> While not directly addressed in this work, many recent improvements in generative AI find their origin in non-equilibrium or stochastic thermodynamics. Diffusion models and the recent focus on annealed importance sampling are prominent examples, with original papers [5, 6] directly relating them to the Jarzynski equality. As such, we do not see it as inconceivable that introducing the wider machine learning community at ICLR to other literature in this domain (Crooks fluctuation theorem, MBAR, operator splitting) will have a potential positive impact on further developments in the domain of generative AI.
>
> **Regarding the guarantees:**
>
> This is an interesting topic. Theorem 2.3 on page 4 addresses this question to some extent; if the time-dependent density and the escorting force collectively solve the continuity equation, the escorting work for a single trajectory will always equal the free-energy difference. We will highlight this point further as a special scenario.
>
> **Regarding the flow architecture:**
>
> The reviewer is correct here; the flow architecture used is an off-the-shelf approach. We do believe that by considering other, more modern approaches the method can be improved, but decided against doing this as we believe that this would unfortunately take away from showing the methodological innovation with respect to the free-energy estimation problem.
>
> ----
>
> We again thank the reviewer for their positive response. We will address their concerns in text and notify the reviewer when this is completed.
>
> **References:**
>
> [1] Huguet, Guillaume, et al. "Sequence-augmented SE (3)-flow matching for conditional protein generation." Advances in neural information processing systems 37 (2024): 33007-33036.
>
> [2] Corso, Gabriele, et al. "DiffDock: Diffusion Steps, Twists, and Turns for Molecular Docking." The Eleventh International Conference on Learning Representations.
>
> [3] Holdijk, Lars, et al. "Stochastic optimal control for collective variable free sampling of molecular transition paths." Advances in Neural Information Processing Systems 36 (2023): 79540-79556.
>
> [4] Du, Yuanqi, et al. "Doob's Lagrangian: A Sample-Efficient Variational Approach to Transition Path Sampling." Advances in Neural Information Processing Systems 37 (2024): 65791-65822.
>
> [5] Sohl-Dickstein, Jascha, et al. "Deep unsupervised learning using nonequilibrium thermodynamics." International conference on machine learning. pmlr, 2015.
>
> [6] Neal, Radford M. "Annealed importance sampling." Statistics and computing 11.2 (2001): 125-139.

---

### Official Review · Reviewer_mX26 · 2025-10-31

**Soundness:** 3
**Presentation:** 3
**Contribution:** 3
**Rating:** 6
**Confidence:** 3

**Summary:**

The paper learns escorted non-equilibrium protocols for free-energy estimation by pairing Conditional Flow Matching for the escort field with a “Conditional Density Matching” objective for the time-dependent potential so the pair approximately satisfies the continuity equation. It also adds two pragmatic pieces: Lie–Trotter splitting to make work evaluation cheaper and an Escorted Protocol Flow Graph (EPFG) so K states only need K–1 learned protocols but still feed MBAR with all pairwise work values. On alanine dipeptide, the approach improves accuracy over TFEP and benefits from EPFG concatenation.

**Strengths:**

- Overall, well-written paper.
- Jointly learn the escort vector field and the time-dependent potential (CFM + CDM) so the learned pair targets the continuity equation, which is exactly what E-NEQ needs.
- Lie–Trotter splitting reduces the number of divergence evaluations in the work computation—practically important for MD-style forces.
- Train only $K–1$ protocols yet still populate MBAR with all $W_{i\to j}$, cutting training cost while improving multi-state consistency.

**Weaknesses:**

- The empirical validation is limited to ADP; it’s not obvious how the training and stability behave on larger biomolecular benchmarks.
- Some integrator choices remain touchy on long concatenations.
- The CDM/DM construction is essentially an MLE on a time-indexed density $p_\theta(x,t)$; the paper introduces new terminology (“Density Matching”) but the core estimator is maximum likelihood on $(x,t)$ samples (later replaced by conditional sampling). That raises questions about identifiability (many $(b_\phi,U_\theta)$ pairs can satisfy the continuity equation approximately) and about how model misspecification in $p_\theta(x,t)$ propagates into work-estimate bias.

**Questions:**

- How is the density matching (DM) in Equation 15 different from maximum likelihood estimation? (I’m confused by the terminology—why call it “Density Matching” when Eq. (15) is a negative log-likelihood over \((x,t)\)?) In line 263, the authors say “MLE”—that’s maximum likelihood estimation, right (still need to define it!)?
- The method learns both $b_\phi(x,t)$ and $U_\theta(x,t)$ to “collectively” solve the continuity equation. Is there an identifiability or degeneracy issue—i.e., multiple $(b,U)$ pairs giving similar likelihood but very different work statistics? What regularization or constraints help?
- In the Lie–Trotter split, how large can the time step be before the discrete E-NEQ estimator’s bias shows up meaningfully in MBAR? Any diagnostic or adaptivity you recommend during training/evaluation?
- EPFG builds all pairwise work values from K–1 learned edges. Do we lose anything versus learning each pair directly (e.g., variance concentration on hard pairs), and could the graph topology (choice of hub k) be learned or adapted?

---

> ### Author Response · Authors · 2025-11-28
> **Initial response by authors**
>
> We thank the reviewer for their positive review of our paper. We are happy to hear that the reviewer found our paper to be well written and appreciate the reviewer’s clear questions focused on clarifications and extensions/future work considerations. We have addressed them below and look forward to the reviewer’s further input.
>
> **Regarding the experimental evaluation:**
>
> We agree with the reviewer that the study of the presented approach on larger biomolecular benchmarks is an important direction for future work. However, as scaling beyond simpler peptide systems primarily becomes more of an engineering rather than a methodological question, we have left it out of this initial proposed method. We would also like to note that the scale of the presented empirical evaluation is in line with those considered broadly in the computational chemistry field for new methodologies, as also highlighted by Reviewer X5Rb.
>
> **Regarding the difference between density matching and MLE:**
>
> We thank the reviewer for bringing up this point, as this should be made more clear in the paper and the abbreviation clarified. Density Matching is essentially maximum likelihood estimation (MLE), with the additional component that the density is time dependent. The framing as “Density Matching” was chosen to align it with the “Flow Matching” terminology, where the intractability of the full learning objective is resolved by introducing a conditional version that has equivalent gradients, which we believe is a new insight of our work.
>
> We are considering renaming Density Matching to “Time-Dependent MLE” and Conditional Density Matching to “Time-Dependent Conditional MLE” to make the relationship with MLE clear and would appreciate the reviewers input.
>
> **Regarding the continuity equation and identifiability of the solution:**
>
> This is a very interesting comment and we thank the reviewer for bringing it up. As the reviewer correctly identifies, Theorem 2.3 does not provide a single solution. There is, however, a very interesting line of research within the stochastic thermodynamics literature that specifically considers finding escorted protocols with minimal dissipated work along the trajectories. Notable here are [1] and [2].
>
> This line of research was not discussed in the paper due to space constraints and a desire not to overly complicate the work, but it did serve as the main motivation for using optimal transport (OT) couplings for Flow Matching. Using OT couplings enforces the escorting velocity field to closely align with the OT plan, which is known to minimize the dissipated work
>
> Based on the reviewer’s comments, we have added a note on this when discussing the model details in section 5.
>
> **Regarding the Lie-Trotter splitting:**
>
> This is closely related to the point above. By using OT couplings we enforce the velocity field to primarily form straight lines, which subsequently allows for relatively large time steps. We did not consider any concrete diagnostics or adaptivity, but agree that this is an important area to explore further. We will add a comment on this to the Discussion.
>
> **Regarding EPFG:**
>
> Learning each pair would be beneficial, as it would add additional datapoints for MBAR to reconstruct the free-energy difference. This does, however, come with additional training costs. The reviewer’s idea to adaptively learn the graph to minimize the number of training steps while minimizing variance would be very interesting to explore. We only considered a graph with a single central node but any connected graph would work.
>
> As part of ongoing work we are currently exploring using the graph during training, instead of learning each pair independently. We thank the reviewer for their insight and will also consider adapting the graph structure itself going forward. We will add this discussion to the Future Work section.
>
> ----
>
> We again thank the reviewer for their positive response. We will address their concerns in the manuscript and notify the reviewer when this is completed.
>
> **References:**
>
> [1] Chen, Yongxin, Tryphon T. Georgiou, and Allen Tannenbaum. "Stochastic control and nonequilibrium thermodynamics: Fundamental limits." IEEE transactions on automatic control 65.7 (2019): 2979-2991.
>
> [2] Zhong, Adrianne, and Michael R. DeWeese. "Beyond linear response: Equivalence between thermodynamic geometry and optimal transport." Physical Review Letters 133.5 (2024): 057102.

---

### Official Review · Reviewer_6Qux · 2025-10-31

**Soundness:** 1
**Presentation:** 2
**Contribution:** 2
**Rating:** 4
**Confidence:** 2

**Summary:**

The authors propose a neural formulation of the Escorted Jarzynski Equality (E-JE) to estimate free-energy differences with reduced variance. The authors reinterpret the E-JE framework using neural potentials $U_\theta$ and escort fields $b_\phi$, trained via conditional flow matching to approximate the optimal nonequilibrium protocol. The paper is well written, with clear theoretical exposition and solid mathematical grounding, providing an intuitive bridge between nonequilibrium statistical mechanics and neural flow–based modeling. Experiments on the alanine dipeptide system demonstrate the feasibility of the approach and compare its performance with classical methods such as Umbrella Sampling and TFEP.

**Strengths:**

1. Provides clear and pedagogical background on the Jarzynski Equality (JE) and its escorted extension, making a technically challenging topic highly accessible.
2. Builds on a solid theoretical foundation, with mathematically consistent derivations and well-motivated neural extensions.
3. The formulation elegantly connects non-equilibrium statistical mechanics with modern neural flow modeling, offering a promising bridge between physics-based and data-driven approaches.

**Weaknesses:**

**Limited discussion of related work and positioning**
The paper presents a neural formulation of the Escorted Jarzynski Equality (E-JE), but does not clearly situate it within the broader landscape of neural flow, Schrödinger bridge, or optimal control–based free-energy estimation methods. Moreover, the boundary between background explanation and related work discussion is somewhat blurred, making it difficult to discern which parts of the formulation are newly contributed by this paper versus prior theoretical developments. A clearer distinction between foundational background and the paper’s novel contributions would help readers better appreciate the originality of the work.

**Lack of verification for continuity-equation consistency**
A central theoretical condition of the proposed method is the continuity equation
$$\partial_t p_\theta + \nabla \cdot(p_\theta b_\phi)=0,$$
ensuring that $U_\theta$ and $b_\phi$ form a consistent escorted protocol. However, the paper does not quantitatively evaluate or visualize how well this condition is satisfied after training.

**Missing JE ($b$ = 0) baseline — variance reduction unquantified**
The main purpose of E-JE is to reduce variance relative to the standard JE, but the JE ($b$ = 0) baseline is entirely omitted from experiments.
Although both yield identical $\Delta F$ values, without reporting $\mathrm{Var}(e^{-\beta W})$ it is impossible to demonstrate the actual benefit of the learned escort field.

**Limited experimental diversity and missing ablations**
Experiments are restricted to a single system (alanine dipeptide) and comparisons are limited to classical methods such as TFEP and umbrella sampling. There is no evaluation against recent neural baselines or component-wise ablations.

**Insufficient explanation for separating $b_\phi$ and $U_\theta$**
It is unclear why $b_\phi$ (flow field) and $U_\theta$ (potential) must be trained as independent networks rather than being coupled through a gradient or consistency constraint. A brief discussion on the physical or algorithmic motivation for this design choice (e.g., Helmholtz decomposition or non-conservative flow representation) would improve clarity.

**Questions:**

1. **JE baseline and variance reduction**
Since the standard Jarzynski Equality (JE) corresponds to the case ($b$ = 0), could you include this baseline to quantify the actual variance reduction achieved by the learned escort field?

**Details Of Ethics Concerns:**

During this review, LLM was used to assist in surveying the theoretical background.

---

> ### Author Response · Authors · 2025-11-28
> **Initial response by authors**
>
> We thank the reviewer for their comments. We are glad to hear that the reviewer found our work to provide a clear pedagogical background, the mathematical framework well formulated, and the proposed solution promising. A large focus of our work was to clearly establish the connection between the physics-based and data-driven approaches, and based on the reviewer’s comments we are glad to hear that this has succeeded. We hope that with this we have not only provided a first proposal for learning escorted protocols for multistate free-energy estimation, but also a starting point for follow-up work.
>
> We have addressed the concerns of the reviewer below. We hope that, with the discussion below as well as the ongoing updates to the manuscript, the reviewer feels confident to increase their score.
>
> **Regarding related work and positioning:**
>
> We would like to thank the reviewer for this note and agree that this part of the paper can be improved. We are adding an additional dedicated “Related Work” section between the methodology and the discussion section. This section focuses on the topics that the reviewer highlights; specifically, it has two paragraphs on:
>
> 1. Other traditional methods for free-energy estimation such as thermodynamic integration, and transition path sampling.
>
> 2. Other neural methods for free-energy estimation and how our method is different. This discussion will specifically focus on related path-based free-energy estimation approaches such as TFEP and thermodynamic integration. This primarily extends on the short discussion already included in Lines 391–394.
>
> We will also add a dedicated paragraph to the Introduction to more clearly state the contributions of our work.
>
> **Regarding continuity-equation consistency and separation of the learning flow field and potential:**
>
> Grouping these two topics together, we agree with the reviewer that consistency between the velocity field and the potential is an important aspect of reducing the dissipated work and thus the variance of the free-energy difference estimator. In the work presented here we have focused on highlighting that it is feasible to learn both components (flow field and density) needed for escorted free-energy estimation, but do not want to claim that this is the best possible method to do so.
>
> While the current framework (using the same conditional density and coupling for the Conditional Flow Matching and Conditional Density Matching objectives) provides a learning signal that enforces consistency between $U$ and $b$, we agree that exploring hard constraints on this would be valuable for future work.
>
> We do want to note that this is, however, not trivial to implement as a single model due to the escorting velocity field not corresponding to any specific component of the time-dependent density or its gradient. We will add this clarification, as well as the note on future work, to the paper.
>
> **Regarding missing JE baseline:**
>
> We understand the reviewer’s concern and would like to inform the reviewer that we are currently adding the JE ($b=0$) baseline to the work. We consider for this purpose both the scenario where the same number of timesteps are used as in the E-JE and TFEP experiments, as well as a version in which the number of timesteps is increased for the system to behave close to quasi-static. Preliminary results show the high timestep requirement for the JE baseline to achieve sufficiently low variance.
>
> **Regarding experimental diversity:**
>
> We would like to acknowledge the reviewer’s concern and thank them for raising it. To clarify, Targeted Free-Energy Perturbation (TFEP) as presented in the experimental section serves as a neural free-energy estimation baseline. As very briefly discussed in Lines 391–394, TFEP is currently the only protocol-based traditional free-energy estimation approach that is widely considered in the machine learning community, resulting in multiple papers discussing possible directions. The version of TFEP we compare our work against (implemented by using only the same learned velocity field as used for our E-NEQ method) falls within this category of neural estimators.
>
> ---
>
> We again thank the reviewer for their considerate response. We will address their concerns in the manuscript and notify the reviewer when this is completed. We look forward to hearing from the reviewer and are happy to make any further clarifications and adjustments where needed.

---

### Official Review · Reviewer_k7x6 · 2025-10-31

**Soundness:** 2
**Presentation:** 1
**Contribution:** 2
**Rating:** 2
**Confidence:** 3

**Summary:**

This paper proposes a method for learning escorted switching protocols to estimate free-energy differences between multiple thermodynamic states. The approach combines Conditional Flow Matching (CFM) to learn a deterministic escorting vector field and introduces Conditional Density Matching (CDM) to learn a time-dependent potential. The authors also propose Lie-Trotter splitting to reduce computational cost and a flow graph construction to scale to multiple states. Experiments on alanine dipeptide (ADP) show improvements over Targeted Free-Energy Perturbation (TFEP) in terms of estimation accuracy.

**Strengths:**

- The paper tackles an important problem in computational biochemistry: efficient and accurate free-energy estimation across multiple states.

- The idea of jointly learning the escorting vector field and the time-dependent potential via CFM and CDM is a meaningful attempt to reduce variance in free-energy estimators.

- The use of Lie-Trotter splitting and flow graph construction addresses practical challenges in computational cost and scalability.

- Experimental results on ADP show that the proposed E-NEQ estimator outperforms TFEP in several settings.

**Weaknesses:**

Overall, it is hard for me to follow this paper, and I think there are many related but not well discussed in this work. But I would be open to adjusting my score if the authors can address my concerns.

- It is very hard for me to identify what the actual contribution of this paper is.
  - If I understand the work correctly, Sections 2 and 3 both appear to be preliminary material rather than novel methods.
  - Section 4 supposedly presents the proposed algorithm, but the description is vague and lacks a clear algorithmic definition. I would recommend a pseudo-code to present the algorithm clearly.
  - It is unclear to me what exactly is being optimized or learned, and what the goal of this paper is to address.
- The methodology is vague to me.
  - Section 3 mentions flow matching but does not clearly define the forward and backward processes, how the reverse flow is parameterized, or how training is performed.
  - Section 4 seems to involve a neural network trained with ideas similar to parallel tempering, but no pseudo-code or mathematical formulation is provided.
  - Section 5 discusses Overdamped & Underdamped Langevin and Hamiltonian Monte Carlo, which makes the algorithm a bit clearer. But it is unclear to show the continuous-time SDEs, their discretization schemes, or how stochastic gradients are estimated from data.
  - The use of Metropolis–Hastings correction seems questionable given the high computational cost in high-dimensional generative models, yet the paper does not justify this design choice.
- There is no dedicated related work section, and the paper does not discuss how it differs from prior research in replica exchange or parallel tempering methods. To me, the approach appears very similar in spirit to parallel tempering/replica exchange MCMC methods [1–4], which also sample across multiple states to balance exploration and exploitation. The paper does not clarify whether the proposed method offers any theoretical or empirical advantage over the related work.
- There are some undefined notations or some presentation issues. If I missed some, please let me know where they are. For example, I did not find some important quantities such as $\alpha_R$, $\alpha_L$, $\alpha_D$, $\beta$, $C_5$, and $\alpha'$ in Table 2, and variables $\phi$ and $\psi$ in Figure 4. This makes the results hard to interpret for readers unfamiliar with Alanine Dipeptide.

References:

[1] Accelerating Nonconvex Learning via Replica Exchange Langevin Diffusion. ICLR 2019

[2] Non-convex Learning via Replica Exchange Stochastic Gradient MCMC. ICML 2020

[3] Spectral Gap of Replica Exchange Langevin Diffusion on Mixture Distributions, Stochastic Processes and their Applications, 2022

[4] Non-reversible Parallel Tempering: A Scalable Highly Parallel MCMC Scheme, Journal of the Royal Statistical Society Series B: Statistical Methodology, 2022

**Questions:**

- Why is the Metropolis–Hastings correction necessary here, and how is it made computationally feasible in high-dimensional settings?
- How are the Langevin dynamics discretized? What step size, friction, and stochastic gradient estimation schemes are used?

---

> ### Author Response · Authors · 2025-11-28
> **Initial response by authors**
>
> We would like to thank the reviewer for their in-depth review. We are glad the reviewer agrees that the problem is of practical importance, and that the proposed approach is promising. We also appreciate the reviewer’s note that they would be open to adjusting their score if the concerns are addressed, and we hope to do so below.
>
> **Regarding the clarity:**
>
> Based on the reviewer’s comments, we believe we can substantially improve the manuscript’s clarity, as the concerns appear primarily driven by insufficient exposition. We are currently making the following edits:
> - Add a dedicated section in the Introduction summarizing the core contributions (also requested by Reviewer 6Qux)
> - Add a clearer forward reference to the Appendix where formal metastable-state definitions are given, and include an illustration of the alanine dipeptide dihedral angles $\phi$ and $\psi$.
>
> Regarding “stochastic gradients estimated from data,” it is not fully clear to us which part of the paper suggested stochastic-gradient estimation from data. Our proposed method uses the gradient of the learned density which is given by backprop through the network. We are happy to clarify this in the paper and would appreciate the reviewers input on which area of the paper caused the confusion.
>
> **Regarding the contribution:**
>
> We thank the reviewer for raising concerns regarding the contribution of our work. We hope that, with the comments below and the adjustments currently being made to the manuscript, these concerns can be alleviated.
>
> The reviewer is correct that Sections 2 and 3 primarily focus on preliminary material. Section 2 is used to provide an in-depth overview of non-equilibrium free-energy estimation and place it into a notation familiar to the machine learning community. While not novel, we do see this as an important contribution to make the field more accessible. Section 3 then directly relates the required components discussed in Section 2 into the popular Flow Matching framework. While Flow Matching is well established, our proposed extension (Density Matching) specific to the escorted free-energy estimation problem considered here, is novel.
>
> The primary learning objectives used in the paper are therefore the two loss terms discussed in Section 3 (Eqs. 14 and 16). These respectively learn the time-dependent vector field and density required to construct the escorted protocol discussed in Section 2 (Def. 2.2).
>
> Section 4 then introduces two concepts (Lie–Trotter splitting and the Escorted Flow Graph) that are used to make the calculation of the required work values sufficiently efficient in the multistate setting.
>
> **Regarding related work, notably replica exchange and parallel tempering:**
>
> The reviewer is correct in pointing out Parallel Tempering (PT) and Replica Exchange (RE) as a closely related concepts that originate from the molecular dynamics field and have found adaptation within the machine learning community. Both PT and RE are built upon the idea of defining multiple thermodynamic states that are explored in parallel. This is similar to how the switching protocol used in non-equilibrium methods defines an interpolation between different thermodynamic metastable states. Both approaches are also similar in that they are often combined with MBAR to combine more than two thermodynamic states.
>
> There is, however, a clear distinction between the use of PT and RE and the line of research discussed in our work. Where PT and RE are methods that exploit pre-defined thermodynamic states to improve exploration, our work proposes a framework to learn the optimal escorted switching protocol that interpolates between different thermodynamic states.
>
> Another contrast between the general problem setting considered in our work (free-energy estimation) and the domains in which parallel tempering and replica exchange are often considered is that free-energy difference estimation focuses on estimating a difference between distinct thermodynamic states. PT and RE are, on the other hand, commonly used to estimate a quantity for a single thermodynamic state by constructing auxiliary states to improve exploration.
>
> As discussed in our response to Reviewer 6Qux, we are adding a related work section after Section 5.

---

> > ### Author Response · Authors · 2025-11-28
> > **Initial response by authors (2)**
> >
> > **Regarding Metropolis-Hastings and general integration setup:**
> >
> > One core restriction in the use of the Jarzynski equality and its extensions is that the time-dependent dynamics have the learned density as their stationary distribution. While continuous-time SDEs in the form of Langevin and Hamiltonian dynamics satisfy this criterion, this is not true for their discretized versions. Using Metropolis–Hastings is therefore a formally required step not just for our method, but for non-equilibrium sampling frameworks more broadly. In practice, it is often dropped in large-scale molecular dynamics simulations.
> >
> > We opted to keep the Metropolis–Hastings step to remain close to the theory. However, for future work, we strongly believe that it is not necessary to keep this step, and instead to align the type of integrators used more closely with those already used in molecular dynamics. We thank the reviewer for this comment and will make sure to highlight why we use Metropolis–Hastings here, and why we believe it can be dropped in our ongoing update.
> >
> > ---
> >
> > We would like to again thank the reviewer for their considerate response. We will address their concerns in the manuscript and notify the reviewer when this is completed. We look forward to hearing from the reviewer and are happy to make any further clarifications and adjustments where needed.

---

### Meta-Review · Area_Chair_Z69j · 2025-12-30

**Summary:**

Reviewer k7x6 mainly concerned about the readability of the manuscript and the missing discussion on related work.

Reviewer 6Qux primarily concerned about the limited empirical evaluation and discussion on related work.

As for the contribution, I believe the proposed method is a valid improvement upon directly leveraging flow matching techniques to two-state free-energy estimation problem.

**Reviewer Concerns:**

I believe readability concerns mainly comes from the significant amount of space used to introduce the physical background of the problem. It is indeed necessary for audience without enough background on the thermodynamics to understand the problem setup.

The authors also provide an additional section to discuss the related work, which I think is sufficient to address the concern of the reviewers.

**Reviewer Scores:**

Reviewer k7x6 and 6Qux may improve their score as authors' rebuttal and update of the manuscript resolves most of their concerns.

---

### Decision · Program_Chairs · 2026-01-26

Accept (Poster)